# Non-Isolated Neural Tube Defects with Comorbid Malformations Are Responsive to Population-Level Folic Acid Supplementation in Northern China

**DOI:** 10.3390/biology11091371

**Published:** 2022-09-19

**Authors:** Xiaoyu Che, Jufen Liu, Gabriel L Galea, Yali Zhang, Nicholas D. E. Greene, Le Zhang, Lei Jin, Linlin Wang, Aiguo Ren, Zhiwen Li

**Affiliations:** 1Institute of Reproductive and Child Health/Key Laboratory of Reproductive Health, National Health Commission of the People’s Republic of China, Peking University, Beijing 100191, China; 1710306230@pku.edu.cn (X.C.); zhangyl@bjmu.edu.cn (Y.Z.); zhangle@bjmu.edu.cn (L.Z.); jinlei@bjmu.edu.cn (L.J.); linlinwang@bjmu.edu.cn (L.W.); renag@pku.edu.cn (A.R.); 2Department of Epidemiology and Biostatistics, School of Public Health, Peking University, Beijing 100191, China; 3Developmental Biology and Cancer Department, UCL Great Ormond Street Institute of Child Health, University College London, London WC1N 1E, UK; g.galea@ucl.ac.uk (G.L.G.); n.greene@ucl.ac.uk (N.D.E.G.)

**Keywords:** neural tube defects (NTDs), congenital malformation, comorbidity, folic acid, primary prevention, Northern China

## Abstract

**Simple Summary:**

Neural tube defects are severe congenital malformations of the central nervous system. Some cases also have comorbid malformations in other organ systems, which cause morbidity and mortality in affected individuals. Although folic acid is effective in preventing neural tube defects, whether folic acid prevents those cases which have comorbid malformations, or only isolated neural tube defects is unknown. In this study, we described the epidemiology of neural tube defects with comorbid malformations and assessed the impact of folic acid supplementation. We found the prevalence of neural tube defects with comorbid malformations decreased after population-level folic acid supplementation in northern China. Malformations of various organ systems are more common in people with neural tube defects, suggesting common etiology. For fetuses with NTDs, clinicians are also suggested to consider screening for possible comorbid congenital malformations.

**Abstract:**

Objective: Comorbid congenital malformation of multiple organs may indicate a shared genetic/teratogenic causality. Folic acid supplementation reduces the population-level prevalence of isolated neural tube defects (NTDs), but whether complex cases involving independent malformations are also responsive is unknown. We aimed to describe the epidemiology of NTDs with comorbid malformations in a Chinese population and assess the impact of folic acid supplementation. Study Design: Data from five counties in Northern China were obtained between 2002 and 2021 through a population-based birth defects surveillance system. All live births, stillbirths, and terminations because of NTDs at any gestational age were recorded. NTDs were classified as spina bifida, anencephaly, or encephalocele. Isolated NTDs included spina bifida cases with presumed secondary malformations (hydrocephalus, hip dislocation, talipes). Non-isolated NTDs were those with independent concomitant malformations. Results: A total of 296,306 births and 2031 cases of NTDs were recorded from 2002–2021. A total of 4.8% of NTDs (97/2031) had comorbid defects, which primarily affected the abdominal wall (25/97), musculoskeletal system (24/97), central nervous system (22/97), and face (15/97). The relative risk of cleft lip and/or palate, limb reduction defects, hip dislocation, gastroschisis, omphalocele, hydrocephalus, and urogenital system defects was significantly greater in infants with NTDs than in the general population. Population-level folic acid supplementation significantly reduced the prevalence of both isolated and non-isolated NTDs. Conclusion: Epidemiologically, non-isolated NTDs follow similar trends as isolated cases and are responsive to primary prevention by folic acid supplementation. Various clinically-important congenital malformations are over-represented in individuals with NTDs, suggesting a common etiology.

## 1. Introduction

Neural tube defects (NTDs) are severe congenital malformations of the central nervous system (CNS) which affect approximately 1 in every 1000 births globally [1]. Open NTDs, such as spina bifida and anencephaly, are caused by the failure of neural tube closure, which is normally completed at around 30 days of gestation in humans. Some forms of closed NTDs, such as encephalocele, occur after neural tube closure but may be genetically related to open NTDs [2]. While anencephaly is universally fatal, spina bifida and encephalocele are often amenable to surgical correction. Malformations in other organ systems are important causes of morbidity and mortality in individuals who survive with NTDs. Independent, comorbid malformations may reflect pleomorphic effects of genetic/teratogenic insults on the morphogenesis of multiple organ systems.

Some malformations occur secondary to the NTD, such as Chiari II malformation, hydrocephalus, congenital hip dislocation, and talipes in individuals who have spina bifida [3,4,5]. However, animal studies strongly corroborate the potential for the pleiomorphic effects of gene mutations and teratogen exposure on multiple organ systems. For example, the planar cell polarity pathway (PCP pathway) plays an important role in neural tube closure. Mutations in the PCP pathway can cause NTDs, abdominal wall closure defects, and limb reduction defects in mice [6,7]. In addition, *Grhl3^Cre/+^Rac1^Fl/FL^* mutant fetuses exhibit abdominal wall defects in addition to spinal and brain/head defects [2]. Tissue-specific ablation of integrin β1 may lead to epithelial fusion defects, leading to clinically important birth defects, including NTDs and cleft palates [8]. In humans, genetic predispositions contribute to NTD risk, but their penetrance is complex and involve gene/environment interactions. The best-known environmental factor is maternal folate status. This is clearly shown by clinical trials and by the dramatic reduction in NTD prevalence in Northern China from 12% in 2004 before folic acid supplementation, to 3.2% in 2014 after supplementation [9].

Although most epidemiological studies have focused on isolated NTDs, those with comorbidities are also clinically important. In countries where fetal surgery for spina bifida is available, independent comorbid malformations are typically a surgical exclusion criterion [10]. When counselling parents whose pregnancy has been affected by a non-isolated NTD [11], the evidence for a beneficial effect of folic acid is less robust than for isolated NTDs. A previous study in the USA found a progressive reduction in NTD rates between 1992 and 2009, but no significant change in non-isolated NTD rates over the same period [12]. In mice, folic acid does not rescue NTDs caused by mutations in some genes [13], potentially changing the spectrum of associated malformations in supplemented populations.

Population differences might begin to explain variability in previous estimates of non-isolated NTD prevalence. The proportion of NTDs described as having comorbid malformations in community- or hospital-based studies ranges from 9.1% to 66% [14,15,16,17,18,19,20]. Our previous pathological anatomy study showed that 75.8% of NTD cases had additional malformations, of which the majority were musculoskeletal or visceral defects [21]. However, the definition of comorbid malformations is not standardized across studies, with those using sensitive methodologies, such as fetal autopsy, potentially being biased by analyzing more severe terminated or non-viable cases. Additionally, spina bifida in particular is known to cause secondary malformations due to CSF leakage during gestation and musculoskeletal malformations due to denervation.

To our knowledge, few studies have described the population-level epidemiological status of non-isolated NTDs, excluding potentially secondary malformations. Therefore, we aimed to assess the epidemiological characteristics of comorbid malformations associated with NTDs in Northern China over a time period spanning the introduction of folic acid supplementation.

## 2. Study Design

### 2.1. Birth Defects Surveillance

Five counties in Shanxi Province, namely Pingding, Xiyang, Taigu, Zezhou, and Shouyang were included in the population-based birth defects surveillance system in the current study. The system was established in the early 2000s, and more than 20,000 births were recorded each year. All pregnant women who resided in the study area for more than 1 year were monitored. All live births (births of 28 or more complete gestational weeks), all stillbirths of at least 20 weeks gestational age, and pregnancy terminations at any gestational age following the prenatal diagnosis of NTDs were recorded [22]. NTD diagnosis was conducted by local specialists in maternal-fetal medicine and confirmed by pediatricians at Peking University. The study protocol was reviewed and approved by the Institutional Review Board of Peking University.

### 2.2. Case Classification

The present study includes NTD cases observed between 2002 and 2021. In addition to NTDs, there were 20 types of birth defects monitored, including congenital hydrocephalus, cleft palate and/or cleft lip, congenital ear defects, oesophageal atresia/stenosis, anorectal atresia/stenosis, hypospadias, hydronephrosis, club foot, polydactyly, syndactyly, limb reduction defects, omphalocele, gastroschisis, conjoined twins, Down’s syndrome, and congenital heart disease, among others. We also included other types of defects recorded in the database, such as exposed viscera and cystic hygroma.

All NTD cases were classified as isolated or non-isolated. Congenital hydrocephalus, club foot, and congenital dislocation of the hip are considered to be secondary to spina bifida and were, therefore, included in the isolated NTDs cohort, but were considered non-isolated when they occurred with cranial NTDs. Additional defects were classified according to the organ system primarily affected: central nervous system (CNS) defects, craniofacial defects, gastrointestinal system defects, musculoskeletal system defects, abdominal wall defects, urogenital system defects, and others. Due to the possibility of a case having multiple major defects, subgroup categories are not mutually exclusive.

## 3. Statistical Analysis

For NTDs, three types of prevalence were calculated: prevalence of total NTDs, isolated NTDs, and non-isolated NTDs. In calculating these indicators, the denominator remains the same: the total number of all live births. For other associated defects, two types of prevalence were calculated: prevalence of associated defects in the NTD group (the numerator is the number of cases with NTDs and associated defects; the denominator is the number of NTD cases) and prevalence of associated defects in the non-NTD group (the numerator is the number of cases with associated defects but no NTDs; the denominator is the number of live births without NTDs).

Cochran-Armitage Trend Tests were used to analyze the trend of prevalence of total, isolated and non-isolated NTDs. Chi-square tests were used to compare the effects of the residence of the mother, infant sex, gestational weeks and delivery time of isolated and non-isolated NTDs. Chi-square tests were also used to compare the prevalence of isolated and non-isolated NTDs before and after national folic acid supplementation. Poisson test was used to compare the prevalence of associated defects in the NTDs group and non-NTDs group. The types and prevalence of defects associated with NTDs and the prevalence of three subtypes of NTDs combined with other defects were described. Two-tailed *p* < 0.05 was considered statistically significant. All statistical analyses were performed using the R 4.0.2 software.

## 4. Results

### 4.1. The Prevalence, Types and Proportion of Defects Co-Occurring with NTDs

From 2002 to 2021, a total of 296,306 births and 2031 cases of NTDs were recorded in the Shanxi surveillance system. A total of 28.4% (334/1175) of spina bifida cases suffered from defects believed to arise secondary to their NTD, of which 293 had congenital hydrocephalus, 26 cases had clubfoot, and 15 cases had both congenital hydrocephalus and horseshoe foot. These “secondary” co-morbid malformations were included in the isolated NTD grouping.

Additionally, 4.8% of NTDs (97/2031) had co-morbid defects not known to be secondary to spina bifida (Figure 1), including abdominal wall defects (25.8%, 25/97), musculoskeletal system defects (24.7%, 24/97), central nervous system defects (22.7%, 22/97), craniofacial defects (15.5%, 15/97), urogenital system defects (6.2%, 6/97), gastrointestinal system defects (2.1%, 2/97) and other defects (11.3%, 11/97). These 97 infants had 105 associated defects (some infants had defects in more than one site). There were six cases of spina bifida with two additional defects (limb reduction defect and gastroschisis; limb reduction defect and cleft palate; limb reduction defect and reproductive system defect; syndactyly and gastroschisis; cleft lip and palate and polydactyly; anorectal atresia or stenosis and reproductive system defect). There was also a case of multiple NTDs (anencephaly and spina bifida), with a limb reduction defect and diaphragmatic hernia. Another case of multiple NTDs (anencephaly and encephalocele) had cleft lip and palate and clubfoot. In summary, co-morbid malformations in individuals who have NTDs are uncommon but can affect multiple organ systems and have the potential to diminish the quality of life of individuals who survive.

### 4.2. Comorbid Malformations Over-Represented in Individuals with NTDs

We next assessed whether malformations of specific structures are over-represented in individuals who have NTDs, suggesting common genetic or environmental causation. The prevalence of cleft lip or/and palate (*p* < 0.01), limb reduction defects (*p* < 0.01), hip dislocation (*p* < 0.05, excluding cases that also have spina bifida), omphalocele (*p* < 0.01), gastroschisis (*p* < 0.01), hydrocephalus (*p* < 0.05, excluding cases that also have spina bifida), and urogenital system defects (*p* < 0.01) is significantly higher in infants with NTDs than in those born without NTDs (Figure 2).

Associations with the three NTD sub-types were also tested individually (Table 1). Cleft lip and/or palate was more commonly observed in those with spina bifida (*p* < 0.01) or encephalocele (*p* < 0.05) than non-NTD controls. Limb-reduction defects were more common in those with spina bifida (*p* < 0.01) or anencephaly (*p* < 0.01), whereas polydactyly was more common in those with encephalocele (*p* < 0.01). Diaphragmatic hernias were more common in those with anencephaly (*p* < 0.05). Hip dislocation was excluded from the analysis in those with spina bifida but was over-represented in those with encephalocele (*p* < 0.01). Similarly, hydrocephalus was excluded from the analysis of those with spina bifida but was over-represented in those with sub-total anencephaly (*p* < 0.05) and encephalocele (*p* < 0.01). Omphalocele and gastroschisis were over-represented in those with either spina bifida (*p* < 0.01) or anencephaly (*p* < 0.01). Urogenital defects were specifically over-represented in those with spina bifida (*p* < 0.01).

### 4.3. Epidemiological Characteristics of Isolated and Non-Isolated NTD Cases

No differences in the distribution of maternal residence, infant sex, and gestational weeks were observed between the isolated NTD group and those with NTDs and co-morbid malformations (Table 2). Individuals with encephalocele tended to be more likely to have additional malformations.

### 4.4. Temporal Trends in the Prevalence of NTDs

We next asked whether non-isolated NTDs follow the same temporal change in prevalence as isolated NTDs, which are the primary target of prevention programs. The prevalence of total, isolated, non-isolated NTDs, and multiple NTDs all decreased significantly from 115.8/10,000, 109.7/10,000, 6.1/10,000, and 22.3/10,000 in 2003 to 15.5/10,000, 14.3/10,000, 1.2/10,000, and 2.4/10,000 in 2021, respectively (Figure 3A). The 3-year rolling average curves indicate progressively diminishing prevalence of both isolated and non-isolated NTDs (Figure 3B,C). The Cochran–Armitage Trend Test results showed that the prevalence of total NTDs, isolated NTDs, non-isolated NTDs, and multiple NTDs decreased significantly over the past two decades (*p* < 0.01). Both isolated and non-isolated NTD subtypes all showed a significant downward trend, except for non-isolated spina bifida for which this did not reach significance (*p* > 0.05).

### 4.5. Prevalence of Isolated and Non-isolated NTDs before and after National Folic Acid Supplementation

The population-surveillance period included various changes in public policy. The prevalence of non-isolated NTDs decreased from 5.14/10,000 live births in 2002–2008 (before population-level folic acid supplementation) to 2.85/10,000 in 2009–2015 (after population-level folic acid supplementation). China implemented a universal two-child policy in 2016, a family planning policy, which stipulates that eligible couples are allowed to have two children. Then the prevalence of non-isolated NTDs dropped to 1.51/10,000 in 2016–2021 (Figure 4).

After national folic acid supplementation, the prevalence of all NTD groupings, including both isolated cases and those with co-morbid malformations, decreased significantly (*p* < 0.05) (Table 3). Non-isolated cases accounted for 4.7% of NTDs before folic acid supplementation and 4.9% of NTDs after supplementation.

## 5. Discussion

NTDs remain prevalent and clinically important globally. Surgical advances have tremendously improved outcomes for individuals who have spina bifida, but the priority remains population-wide primary prevention. Cases which occur despite adequate maternal folate status are often assumed to reflect genetic/teratogenic causes not responsive to fortification or supplements. Using population-based surveillance data, this study reveals that 4.8% of NTDs have gross comorbid malformations, but that the epidemiology of non-isolated NTDs is comparable to isolated cases, including a reduction in prevalence following population-wide folic acid supplementation.

Various authors have previously suggested that NTDs may be mechanistically linked to other malformations. Czeizel et al. [23] hypothesized the association with other defects of embryonic closure events, including cleft lip or/and cleft palate, posterior cleft palate, diaphragmatic hernia, and omphalocele, each being present in 1–6% of NTD cases. Opitz et al. [24] suggested that midline structures share a particular common developmental vulnerability. Changes in the central nervous system may affect the lip and palate, diaphragm, heart, abdominal wall, and genitalia that share the same midline. Although previous studies [16,17] provide no evidence to support the above two hypotheses, many of the conditions proposed are associated with NTDs, as in our study population. Compared with non-NTD controls, infants with NTDs were more likely to have cleft lip and/or palate, limb reduction defects, hip dislocation, gastroschisis, omphalocele, hydrocephalus, and urogenital system defects.

These associations could be either genetic or teratogenic, if teratogen exposure spans the relevant susceptibility periods. For example, fetal valproate syndrome involves limb reduction defects as well as spina bifida. Genetic causes can also be shared between NTD subtypes and comorbid malformations; Grhl3^Cre^ deletion of Rac1 can cause abdominal call defects, spina bifida, exencephaly, and/or encephalocele in mice [25]. Encephalocele etiology is poorly understood and future work will be needed to explain our observation that they account for a larger proportion of non-isolated NTD cases.

Our results showed that there are no significant differences between isolated and non-isolated NTDs in key demographic characteristics, including the residence of the mother, infant sex, subtype of NTDs, gestational weeks, and delivery time, which suggests that non-isolated NTDs are more likely to be caused by multiple genetic factors. Due to the serious consequences of non-isolated NTDs, we need to further clarify the cause and identify other ways to reduce the risk of non-isolated NTDs. Further prospective cohort or case-control studies are needed to detect the influences of more factors, such as maternal diseases, childbirth history, and family history.

An unanswered question is whether folic acid prevents both NTDs and their comorbid malformations, or only the NTDs. Our study showed that the prevalence of all types of NTDs decreased significantly after national folic acid supplementation. A recent study in China also showed that periconceptional folic acid use prevents both rare and common NTDs [26]. In our previous study, we found that periconceptional folic acid supplementation can prevent congenital limb reduction defects in people with extremely low folate concentrations in Northern China [27]. Some previous studies showed that perinatal use of multivitamin supplements effectively reduced the risk of multiple birth defects, even after excluding NTDs [12]. However, several studies have shown that folic acid supplementation can reduce the risk of NTDs, but cannot reduce the risk of other birth defects [28].

The prevalence of total NTDs, isolated NTDs, and non-isolated NTDs in China showed a downward trend, which also reflects the remarkable results of preventing and controlling NTDs in the five counties of Shanxi Province in the last 20 years. However, when we conducted trend tests according to different subtypes, the prevalence of non-isolated spina bifida did not show a downward trend. A previous study in South Carolina showed that the prevalence of total NTDs and isolated NTDs decreased significantly from 1992 to 2009, but the prevalence of non-isolated NTDs did not change significantly [29]. It is worth noting that since the three different subtypes of NTDs may be caused by different genetic factors, it is necessary to analyze the epidemiological trends according to different subtypes. Therefore, our results suggest that there may be etiological heterogeneity between non-isolated spina bifida and isolated spina bifida. A previous epidemiological study of NTDs also showed that isolated NTDs were more sensitive to environmental factors, whereas non-isolated NTDs were sensitive to both environmental and genetic factors [30]. In addition, some biomarkers are also considered to play an important role in NTD screening during pregnancy, such as maternal serum alpha-fetoprotein (MSAFP) [31] and proprotein convertase subtilisin/kexin type 9 (PCSK9) [32]. However, whether the same biomarkers are equally predictive for screening isolated NTDs and non-isolated NTDs needs further research. In summary, the etiology of non-isolated NTDs needs to be confirmed by more population studies and laboratory studies.

Our current study found that changes in healthcare policy had an impact on the prevalence of non-isolated NTDs. China implemented the folic acid supplementation policy in 2009 and, compared with 2002–2008, the prevalence of non-isolated NTDs decreased in 2009–2015. After the implementation of the universal two-child policy in 2016, the proportion of older and high-risk pregnant women may have increased, potentially resulting in an increased risk of NTDs [33,34]. Other studies have found the opposite association, with a high birth prevalence of spina bifida among younger mothers [35]. Our study showed that after the implementation of the universal two-child policy, the prevalence of non-isolated NTDs continued to decline. We speculate that there are two possible reasons. First, population-level folic acid supplementation can effectively prevent non-isolated NTDs, so the universal two-child policy had little impact on the prevalence of NTDs. Secondly, the impact of the two-child policy on the overall population is not yet clear. We did not observe a rising trend in the birth population after 2016 in the five counties of Shanxi Province. It is worth noting that the universal three-child policy was implemented in China in 2021, which means that further research on birth defects is needed to evaluate any impact on health outcomes in pregnancy.

Our study has several strengths. First, the surveillance is population-based and used quality control to ensure data quality. Our previous retrospective study showed that the surveillance covered 95.6% of births [36]. Second, prior to the present study, only a few studies have reported the epidemiological characteristics of isolated and non-isolated NTDs in China. We provided more detailed information about associated defects.

Despite the clear strengths of our study, some limitations should be acknowledged. First, the study included data from only five counties, and the results were not representative of other provinces or countries. Second, this study did not collect information on demographic and socioeconomic factors, which prevented us from exploring the risk factors associated with isolated and non-isolated NTDs. Third, the monitoring system mainly reported external defects, so we may have missed some internal defects.

## 6. Conclusions

Epidemiologically, non-isolated NTDs follow similar trends as isolated cases and are responsive to primary prevention by folic acid supplementation. Various clinically-important congenital malformations are over-represented in individuals with NTDs, suggesting common etiology.

## Figures and Tables

**Figure 1 biology-11-01371-f001:**
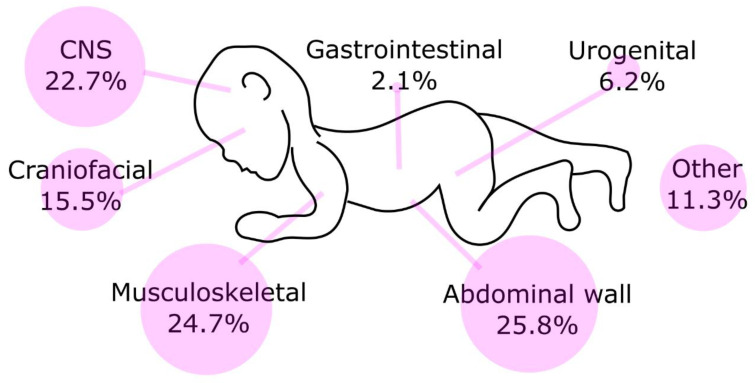
Associated defects among cases with NTDs in five counties in Shanxi province of Northern China, 2002–2021.

**Figure 2 biology-11-01371-f002:**
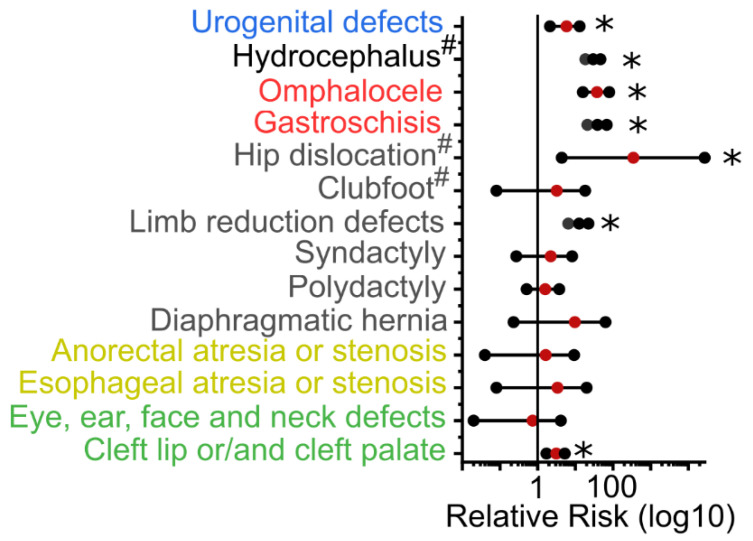
Relative risk of associated defects in NTDs group and non-NTDs group. ^#^ excludes cases with spina bifida. Colors group malformations related to urogenital (blue), CNS (black), body wall (red), musculoskeletal (grey), gastrointestinal (olive) and craniofacial (green) anatomy.

**Figure 3 biology-11-01371-f003:**
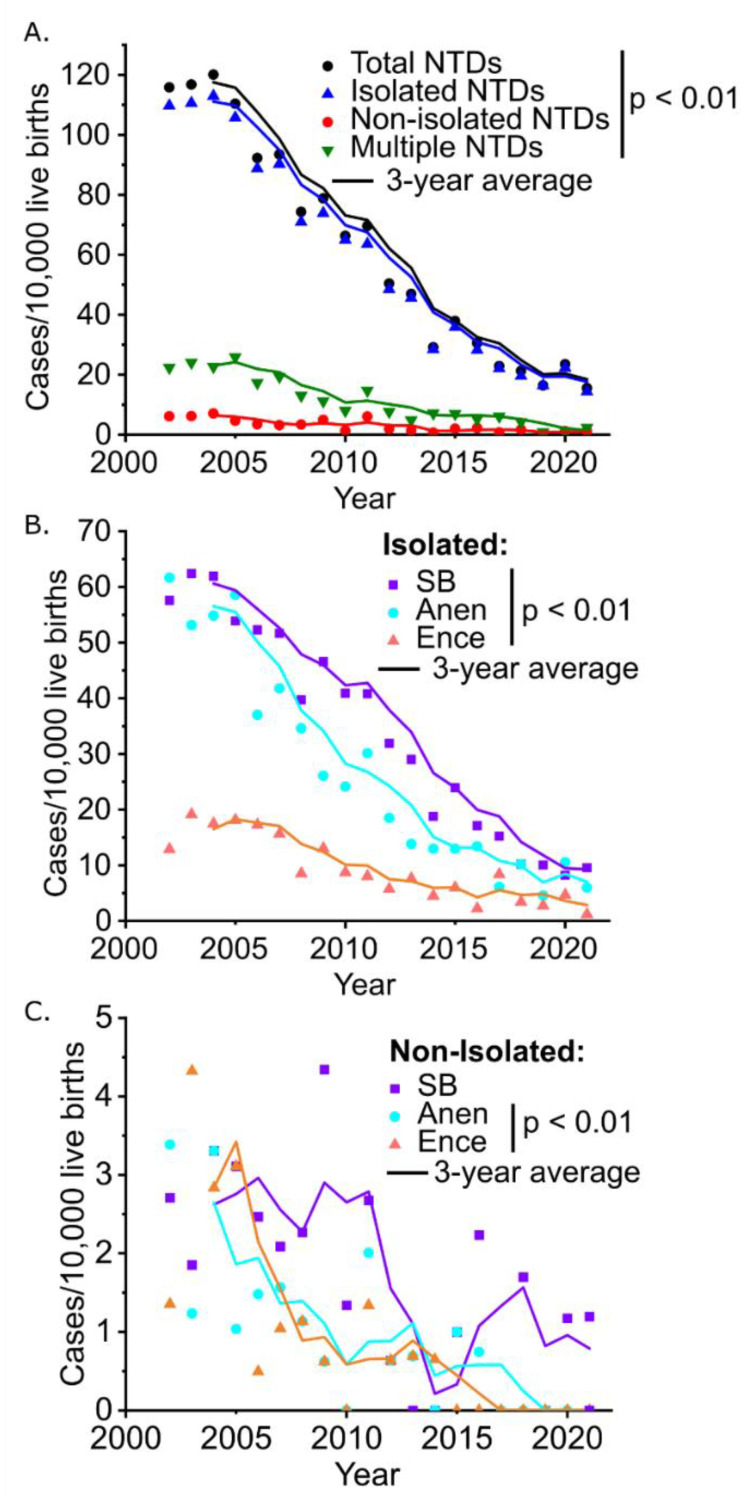
Time trends in the prevalence of NTDs in five counties in Shanxi province of Northern China, 2002–2021. (**A**). Total NTDs; (**B**). Isolated NTDs; (**C**). Non-isolated NTDs. In 2002, only the data of Zezhou, Pingding, and Taigu are presented; In 2003, the data of Zezhou, Pingding, Taigu, and Shouyang are presented; In 2004–2021, the data of Zezhou, Pingding, Taigu, Shouyang, and Xiyang are all presented.

**Figure 4 biology-11-01371-f004:**
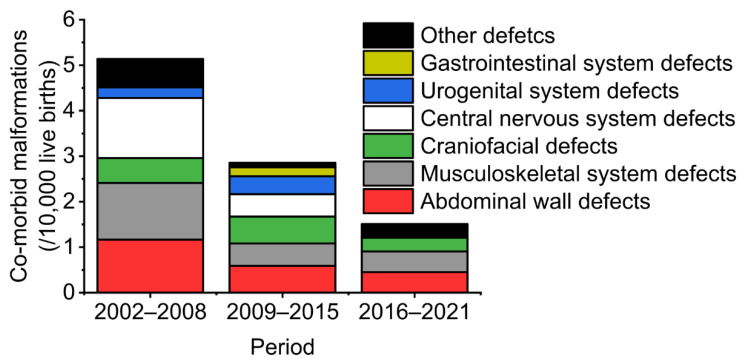
Prevalence of defects co-occurring with NTDs by period in five counties in Shanxi province of northern China, 2002–2021. 2002–2008 is the period before population-level folic acid supplementation; 2009–2015 is the period after population-level folic acid supplementation; 2016–2021 is the period of population-level folic acid supplementation and the universal two-child policy; after 31 May 2021 is the period of population-level folic acid supplementation and the universal three-child policy.

**Table 1 biology-11-01371-t001:** Associated defects in three NTDs by subtype groups and non-NTD group in five counties in Shanxi province of Northern China, 2002–2021.

Birth Defects	Spina Bifida (n = 1175)	Anencephaly (n = 916)	Encephalocele (n = 339)	All NTDs (n = 2031)	Non-NTDs (n = 294,275)
Craniofacial defects					
Cleft lip or/and cleft palate	9 ^†^	5	3 ^&^	14 ^※^	650 ^†&※^
Eye, ear, face and neck defects	1	0	0	1	199
Gastrointestinal system defects					
Esophageal atresia or stenosis	1	0	0	1	43
Anorectal atresia or stenosis	1	0	0	1	89
Musculoskeletal system defects					
Diaphragmatic hernia	1	1 ^#^	0	1	15 ^#^
Polydactyly	3	0	4 ^∮^	5	459 ^∮^
Syndactyly	2	0	0	2	131
Limb reduction defects	12 ^†^	4 ^‡^	0	13 ^※^	150 ^†‡※^
Clubfoot *	/	0	1	1	107
Hip dislocation *	/	0	1 ^∮^	1 ^※^	1 ^∮※^
Abdominal wall defects					
Gastroschisis	13 ^†^	9 ^‡^	0	16 ^※^	60 ^†‡※^
Omphalocele	7 ^†^	4 ^‡^	0	9 ^※^	35 ^†‡※^
Central nervous system defects					
Hydrocephalus *	/	3 ^#^	19 ^∮^	22 ^※^	253 ^#^^∮※^
Urogenital system defects	5 ^†^	1	0	6 ^※^	149 ^†※^

Comparison of prevalence of associated defects between spina bifida group and non-NTD group, ^†^ *p* < 0.01; comparison of prevalence of associated defects between anencephaly group and non-NTDs group, ^#^
*p* < 0.05, ^‡^ *p* < 0.01; comparison of prevalence of associated defects between encephalocele group and non-NTDs group, ^&^
*p* < 0.05, ^∮^ *p* < 0.01; comparison of prevalence of associated defects between total NTDs group and non-NTDs group, ^※^
*p* < 0.01; / excluded defects secondary to spina bifida; * Excluding cases with spina bifida, anencephaly (n = 623), encephalocele (n = 258), all NTDs (n = 856).

**Table 2 biology-11-01371-t002:** Characteristics of isolated and non-isolated NTD cases in five counties in Shanxi province of Northern China, 2002–2021, N (%).

Characteristics		Isolated NTDs (n = 1934)	Non-Isolated NTDs (n = 97)	χ2	P
Residence	Rural	1694 (87.6)	83 (85.6)	0.346	0.556
	Urban	240 (12.4)	14 (14.4)		
Infant Sex	Male	865 (44.7)	40(41.2)	0.021	0.885
	Female	1027 (53.1)	46 (47.4)		
	Unknown	42 (2.2)	11 (11.3)		
Gestational weeks	<28w	1260 (65.1)	61 (62.9)	0.208	0.648
	≥28w	674 (34.9)	36 (37.1)		
NTDs by type	Anencephaly	585 (30.2)	13 (13.4)	34.068	<0.01
	Spina bifida	787 (40.7)	35 (36.1)		
	Encephalocele	206 (10.7)	27 (27.8)		
	Multiple NTDs	356 (18.4)	22 (22.7)		
Delivery time	Before 2009	1262 (65.3)	62 (63.9)	0.073	0.787
	After 2009	672 (34.7)	35 (36.1)		

**Table 3 biology-11-01371-t003:** Prevalence of isolated and non-isolated NTDs before and after national folic acid supplementation in five counties in Shanxi province of Northern China, 2002–2021, N (/10,000).

Type of NTD		Folic Acid Supplementation		χ2	*p*
		Before (n = 128,477)	After (n = 167,829)		
Total NTD	Isolated	1262 (98.2)	672 (40.0)	379.954	<0.01
	Non-isolated	62 (4.8)	35 (2.1)	16.698	<0.01
Spina bifida	Isolated	696 (54.2)	423 (25.2)	162.319	<0.01
	Non-isolated	33 (2.6)	23 (1.4)	5.528	0.019
Anencephaly	Isolated	633 (49.3)	262 (15.6)	273.760	<0.01
	Non-isolated	24 (1.9)	8 (0.5)	13.046	<0.01
Encephalocele	Isolated	202 (15.7)	105 (6.3)	63.003	<0.01
	Non-isolated	26 (2.0)	6 (0.4)	18.709	<0.01

## Data Availability

The data presented in this study are available on request from the corresponding author. The data are not publicly available due to privacy.

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
