# Peer review of "Non-Isolated Neural Tube Defects with Comorbid Malformations Are Responsive to Population-Level Folic Acid Supplementation in Northern China"

_biology, 2022, doi:10.3390/biology11091371_

Round 1

Reviewer 1 Report

Che et al. have aimed to define the epidemiology of neural tube defects (NTDs) with comorbid malformations in a Chinese population and evaluate the impact of folic acid supplementation.  Moreover, they showed that non-isolated NTDs follow similar trends as isolated cases and are responsive to primary prevention by folic acid supplementation epidemiologically. Various clinically-important congenital malformations are over-represented in individuals with NTDs, suggesting common etiology. The study is orginal, well prepared and discused. In my opinion this paper has a potential to make a positive contribution to the co-existing literature. I have only some minor recommendations for the authors:
1- I think that PCSK9 may be involved in the etiopathogenesis of open NTDs at the critical steps of fetal neuronal differentiation. Although it has limitations, PCSK9 may be used as an additional biomarker for the screening of NTDs. The study mentioned below may contribute to this article.

“Erol SA, Tanacan A, Firat Oguz E, Anuk AT, Goncu Ayhan S, Neselioglu S, Sahin D. A comparison of the maternal levels of serum proprotein convertase subtilisin/kexin type 9 in pregnant women with the complication of fetal open neural tube defects. Congenit Anom (Kyoto). 2021 Sep;61(5):169-176. doi: 10.1111/cga.12432. Epub 2021 Jun 23. PMID: 34128273.”

Reviewer 2 Report

General comments: 
- Consider superscripting the references as the numbers interfere with the ease of reading. 
- Several typos or misspelled words. 
- Lines 54-56: consider elaborating
- All the symbols used for Table 1 are a bit confusing and difficult to follow and understand.

Grammar/Spelling Recommendations by line:
42: births
43: "comma" after anencephaly
44: which is normally completed around
56: "period" after mice
57: involve (not involves)
68: "comma" after genes
77: "comma" after methodologies and "comma" after autopsy 
93: remove "apostrophe" from weeks
99: remove the s in NTDs
104: type of defects
125: time OF isolated
129: associated (typo)
137: secondary
202: "comma" after isolated NTDs
202: programs
203: "comma" after 2021
220: surveillance
220: included (typo)
221: prevalence (typo)
223-4: Consider explaining the two-child policy briefly
232: "period" at the end of the sentence
257: Consider adding the word AS to read "associated with NTDs, as in our study population."
270: suggests (not suggested)
274: "comma" after more factors
287: reflects (not reflected)
289: tests
295: suggests (not suggested)
